# Genomic Drivers of Biofilm Formation in *Salmonella* Enteritidis and *S*. Kentucky from Poultry Production

**DOI:** 10.3390/microorganisms13112473

**Published:** 2025-10-29

**Authors:** Jiayi Zhang, Oritsetimeyin Ebosa, Moussa Diarra, Celine Nadon, Tim McAllister, Richard Sparling, Claudia Narvaez-Bravo

**Affiliations:** 1Department of Food and Human Nutritional Sciences, University of Manitoba, Winnipeg, MB R3T 2S2, Canada; zhanga6@myumanitoba.ca (J.Z.); ebosao@myumanitoba.ca (O.E.);; 2Guelph Research and Development Center, Agriculture and Agri-Food Canada (AAFC), Guelph, ON N1G 5C9, Canada; 3National Microbiology Laboratory, Public Health Agency of Canada, Winnipeg, MB R3E 3R2, Canada; 4Department of Medical Microbiology and Infectious Diseases, University of Manitoba, Winnipeg, MB R3E 0J9, Canada; 5Lethbridge Research and Development Centre, Agriculture and Agri-Food Canada, Lethbridge, AB T1J 4B1, Canada; 6Department of Microbiology, University of Manitoba, Winnipeg, MB R3T 2N2, Canada

**Keywords:** biofilm, poultry, *Salmonella*, genomics

## Abstract

*Salmonella* Enteritidis (SE) remains a leading cause of human illness worldwide, and its persistence in poultry environments might be partially attributed to their ability to form biofilm. This study compared the biofilm capacity of 15 SE and 24 *Salmonella* Kentucky (SK) isolates from poultry products and processing facilities to uncover genetic factors driving biofilm heterogeneity. Biofilm formation and curli/cellulose production were evaluated at 20–22 °C. Genomic analyses included phylogenetic reconstruction, comparative system profiling, SNP variation, and BLASTp v2.17.0 comparisons. Phenotypic assays showed that most SE isolates (73%) were strong biofilm formers, while the majority of SK isolates (62%) failed to form biofilms, despite many carrying the complete curli–cellulose gene set and other biofilm-associated genes. Genomic analysis identified 124 biofilm-related genes, 108 of which were conserved across all isolates, and revealed 24 variants with potential functional impact. Mutations in cellulose biosynthesis (*bcs*) genes were linked to weaker biofilms, whereas nonsynonymous variants in *tol* family genes may impair flagellar biosynthesis and matrix stability. These findings demonstrate that genetic variation, not just gene presence, shapes biofilm phenotypes and highlight key molecular targets that may explain why SE persists in poultry production while SK is less successful.

## 1. Introduction

Despite extensive efforts to implement food safety measures and reduce salmonellosis, outbreaks continue to occur, posing a persistent threat to food safety and public health worldwide. Notably, current food safety programs in the United States fell short of achieving the 2020 national public health goal for reducing *Salmonella*-related illnesses [1]. Non-typhoidal *Salmonella* serotypes are one of the leading causes of foodborne illness [2]. *Salmonella enterica* serovar Enteritidis (SE) has been linked to numerous poultry, egg, and egg product-associated outbreaks [3,4,5,6,7]. SE persistence in poultry barns, egg layer barns, and poultry processing facilities remains a persistent problem. In 2023, *Salmonella* Enteritidis is considered the predominant serotype in Canada, accounting for 37.02% of all human salmonellosis, followed by *Salmonella* Typhimurium (6.15%), and *Salmonella* ssp. I 4,[5], 12:I:- (5.46%) [8]. *Salmonella enterica* serovar Kentucky (SK) is another serovar where poultry is the primary carrier, transmitting the bacteria to humans through the consumption of contaminated foods [9]. SK is a polyphyletic non-typhoidal *Salmonella* serovar that comprises two major sequence types (STs), ST152 and ST198, representing distinct genetic lineages [10]. The ST 152 is more prevalent in the U.S. poultry and poultry products. The ST 152 is sporadically linked with human illness; however, it has not been associated with major foodborne *Salmonella* outbreaks in the U.S. [11]. In contrast, ST 198 is more prevalent in international poultry (poultry outside the U.S.) and is more frequently associated with human disease [10]. In the EU, confirmed and reported human salmonellosis cases caused by *Salmonella* Kentucky accounted for only 0.3–0.4% of cases between 2010 and 2016, making it far less prevalent than *Salmonella* Enteritidis, which accounted for 48.0–57.1% during the same period [12]. In 2023, cases of *Salmonella* Kentucky were reported across five Canadian provinces: British Columbia (*n* = 8), Alberta (*n* = 3), Saskatchewan (*n* = 1), Ontario (*n* = 19), and Quebec (*n* = 7) [8].

Similarly to salmonellosis caused by *S.* Enteritidis, human illness from *S.* Kentucky is likely underreported, which may be attributed to its self-limiting nature. However, it is still considered one of the new five target serovars for breeding hens in the EU after 2014, and it had the highest proportion of travel-associated cases (53.2%) in the EU in 2016 [12]. Despite the high prevalence of *Salmonella* Kentucky in poultry and the poultry processing environment, it causes a very low estimated burden of illness; this has been a longstanding conundrum for the food safety and public health communities, and it has been thought to be attributed to low virulence [13].

*Salmonella* continues to evade food safety controls, a persistence that may be linked to its remarkable adaptability and resilience. One mechanism that bacteria use to withstand environmental stress is the formation of biofilms. Biofilms have also been associated with bacterial persistence, primarily due to the extracellular polymeric substance (EPS) matrix, which impedes the penetration of antibiotics and disinfectants. The EPS matrix also provides humidity, which helps bacteria to survive longer [14,15,16]. Biofilm refers to a group of surface-associated microorganisms that can attach to the surface and produce an amorphous extracellular polymeric substance (EPS) matrix. EPS serves as a structural framework for the biofilm’s three-dimensional layout, allowing cells to adhere to surfaces and interact with one another within the biofilm [17]. Bacteria within biofilms possess strong capabilities to survive in hostile environments, as the EPS can protect them against antimicrobial compounds, desiccation, and other environmental stresses [18,19]. The structure of EPS varies based on the ecological conditions in which biofilms are formed. Biofilm EPS matrix consists predominantly of polysaccharides, proteins, phospholipids, and other biomolecules [14]. Biofilms can be composed of single species or multispecies, with either single or multi-layered [20]. Bacteria can form biofilms on numerous surfaces, both biotic and abiotic, and at different temperatures [14]. The ability of *Salmonella* to form resilient biofilms in facilities like barns and processing plants creates a constant source of potential contamination. These persistent isolates can be difficult to eradicate, leading to ongoing food safety hazards and recurring outbreaks. In broiler houses, key risk factors for *Salmonella* contamination include feces, dust, and feed. Day-old chicks and transport vehicles also introduce and recirculate the pathogen. In contrast, the most significant risk factors in laying hen houses are dust, surfaces, and feces [21]. Microbial biofilms readily establish themselves across food-production environments, within water distribution systems and poultry drinkers, as well as on facility surfaces such as floors, walls, ceilings, and equipment (e.g., conveyor belts, plastic, and rubber components) [22]. Bacterial biofilm production can be classified as strong, moderate (or intermediate), weak, or non-biofilm formers [23]. In layer barns in Manitoba, the ambient temperature is carefully controlled and typically maintained within the 20–22 °C range to optimize hen welfare and egg production. However, these temperatures could facilitate *Salmonella* biofilm formation in poultry production environments [24]. The ability of *Salmonella* to establish robust biofilms on diverse surfaces is closely associated with bacterial surface structures, including flagella, fimbriae, and cellulose [25]. The specific type of fimbria called curli, as well as cellulose, are key phenotypic markers commonly associated with the biofilm-forming ability of bacteria. The RDAR morphotype (red, dry, and rough) is a characteristic colony morphology observed in some Enterobacteriaceae family members, particularly S. enterica and *E. coli*, when grown on Congo Red agar. It is commonly associated with biofilm-forming ability and environmental persistence. The RDAR phenotype-expressing colonies consist of proteinaceous compounds, such as adhesive curli fimbriae and exopolysaccharides, including cellulose, as well as an O-antigenic capsule, capsular polysaccharide, and lipopolysaccharide [14]. Curli are amyloid-like cell-surface proteins that are responsible for host colonization, persistence, motility, and invasion in *Salmonella* spp. and have been reported to be essential for biofilm formation since they promote cell-surface and cell-to-cell interactions [14]. By comparing genomic sequences with observed biofilm phenotypes, critical links between a strain’s genetic makeup and its resilience can be revealed. This genomic insight not only enhances our ability to predict the behavior of emerging *Salmonella* strains but also provides the poultry industry with tools for risk assessment and management. In addition, elucidating the genetic basis of strong biofilm formation provides valuable targets for innovative control measures, ultimately supporting the development of more effective strategies to mitigate pathogen persistence and enhance food safety. Accordingly, this research investigated the relationship between isolates of *Salmonella* Enteritidis (SE) and *Salmonella* Kentucky (SK) serotypes exhibiting different biofilm formation phenotypes (strong, intermediate, weak, and non-biofilm formers) and their corresponding genomic profiles. The objective was to identify marker genes linked to biofilm formation that explain variations in bacterial resilience and offer insights for developing targeted control strategies to reduce Salmonella contamination and enhance risk assessment.

## 2. Materials and Methods

### 2.1. Salmonella Biofilm Formation

#### 2.1.1. Salmonella Bacterial Isolates Selection and Culture Conditions

Fifteen *Salmonella* Enteritidis (SE) isolates were included in this study, selected based on their source within the poultry environment, facilities, and processing steps (Table 1). Several SE isolates were recovered from end-products, including chicken nuggets, chicken breast, and chicken feed, indicating their persistence through multiple processing stages. One additional SE isolate (#107) was included as a control, as it was the only intermediate biofilm former that tested negative for curli and cellulose production. In total, all SE isolates were evaluated for their ability to form biofilms. To broaden the range of biofilm phenotypes, 24 *Salmonella* Kentucky (SK) isolates, also isolated from a poultry processing facility in Agassiz, Canada, were included (Table 1). All isolates were obtained from the University of Manitoba culture collection.

#### 2.1.2. Culturing Procedures

All bacterial isolate cultures used in this research were kept in a −80 °C freezer with Trypticase Soy Broth (TSB; Becton, Becton and Dickinson, Sparks, NV, USA) and 15% glycerol. To activate the cultures, a loopful of the frozen culture was streaked onto Xylose Lysine Deoxycholate (XLD) agar plates (Hardy Diagnostics Inc., Santa Maria, CA, USA) and incubated at 37 °C overnight to obtain isolated colonies. A single colony of each bacterial isolate was transferred and inoculated from the overnight agar plate into a 10 mL sterile Lennox Broth no-salt (LN-NS (10 g tryptone (VWR Chemicals LLC, Solon, OH, USA) +5 g yeast extract (Thermo Fisher Scientific, Sasingstoke, UK) in 1 L distilled water and incubated at 37 °C overnight. The overnight culture was adjusted to a concentration of 10^8^ colony-forming units (CFU/mL) by diluting it with LB-NS using a 0.5 McFarland Standard (10^8^ CFU), for culture concentration confirmation. OD (optical density) was measured (0.30 nm to 0.35 nm) on the spectrophotometer (Thermo Scientific™, Genesys 20) for further use.

#### 2.1.3. Biofilm Assessment Using the Crystal Violet Method

For *Salmonella* biofilm assessment, the CV method was followed [26]. For this purpose, *Salmonella* cultures (10^8^ CFU) were diluted to achieve a 10^7^ CFU/mL. In a 96-well microplate, 180 µL of sterile LB-NS was dispensed into each of the 96 wells. Using a multi-channel pipette, 20 µL of each individual *Salmonella* culture was dispensed into the wells to achieve a 10^6^ CFU/mL cells in the microplates. Each plate included one positive control (*E. coli* R508) and one sterility control (200 µL LB-NS with no bacteria). Each experiment was repeated four times. The microplates were incubated at 20–22 °C for 5 days. After the 5-day incubation, biofilm development was assessed. Microplate wells were washed tree times with 300 µL 1.25% Butterfield’s Phosphate Buffer (KH_2_PO_4_) (BPB) (Hardy Diagnostics Inc., Santa Maria, CA, USA) using a microplate washer (405 LS, BioTek, Winooski, VT, USA), then plates were air dried for 10 min, following by adding 200 µL of absolute methanol to each well, for 15 min to fix the attached biofilm. After 15 min, the methanol was removed and air-dried for 15 min. The biofilm was then stained with 200 µL of 0.1% crystal violet (CV) (Sigma Aldrich, St. Louis, MO, USA) [26] for 20 min. Next, the fixed biofilm was washed three times with 200 µL BPB and air-dried for 15 min. Finally, 200 µL of 85% ethanol was added to each well, microplates were agitated for 10 s to dissolve the crystal violet attached to the biofilm. The optical density (OD) was measured at 630 nm using a microplate reader (BioTek ELx800; BioTek Instruments Inc., Winooski, VT, USA).

The data obtained from the microplate reader were analyzed using the method described by Stepanović et al. and Adator et al. [23,26]. The biofilm classification cutoff line was based on the value of the negative controls, where ODc (optical density cut-offs) = mean of negative control+3*SD of the negative control. The biofilm-forming ability, as measured by the OD value, was then adjusted to fit within the range. When OD ≤ ODc, it is a non-biofilm former; when ODc < OD ≤ 2ODc, it is a weak biofilm former; when 2ODc > OD ≤ 4ODc, it is an intermediate biofilm former; when 4ODc < OD, it is a strong biofilm former.

For assessing the biofilm-forming ability of the *Salmonella* isolates, interactions between isolates and biofilm categories were analyzed using the Chi-Square method. For all statistical analyses, significance was declared at *p* < 0.0001.

#### 2.1.4. Curli and Cellulose Detection

*Salmonella* Enteritidis and Kentucky isolates were assessed for their capacity to produce curli and cellulose at 20–22 °C. For curli production, the overnight cultures of *Salmonella* isolates were steaked on Congo red agar (10 g/L casamino acids (VWR, Solon, OH, USA), 1 g/L yeast extract (Thermo Fisher Scientific, Basingstoke, UK), and 20 g/L agar (Becton, Dickinson and Company, Sparks, MD, USA), supplemented with 20 µg/mL Coomassie brilliant blue dye (Sigma Aldrich, St. Louis, MO, USA) and 40 µg/mL Congo red dye (Sigma Aldrich, St. Louis, MO, USA) [26]. For cellulose production, the overnight cultures of *Salmonella* isolates were streaked on Luria–Bertani (LB) agar (Hardy Diagnostics CulGenex^TM^, Santa Maria, CA, USA) supplemented with 200 mg/L Calcofluor dye (Sigma Aldrich, St. Louis, MO, USA). The plates were incubated at 22 °C for 1–2 days. A positive result for curli production is shown as red, dry, and rough (RDAR) morphotype, which indicates the presence of curli in the extracellular matrix; other morphotypes indicate negative results. The presence of fluorescence colonies on LB agar with Calcofluor dye at 366 nm UV light indicates a positive result for cellulose, and colonies that do not fluoresce indicate a negative result.

### 2.2. Genomic Comparison of S. Enteritidis and S. Kentucky with Regard to Their Biofilm Formation Gene Information

#### 2.2.1. Whole Genome Sequence Generation

Genomic DNA from *Salmonella* isolates was extracted using the DNeasy Blood and Tissue Kit-Gram-Negative protocol (Qiagen, Inc., Toronto, ON, Canada) according to the manufacturer’s recommendations. DNA concentration and quality were tested using NanoDrop (NanoDrop Technologies, Wilmington, DE, USA). The whole genome sequence for the fifteen *Salmonella* Enteritidis (SE) isolates and twenty-four *Salmonella* Kentucky (SK) isolates was carried out by the National Microbiological Laboratory, Public Health Agency of Canada, located in Winnipeg.

Quantification of the DNA was performed using the Qubit HS kit. The genomic DNA (gDNA) was first normalized using the Biomek i7 robot. The library samples were prepared using the Illumina DNA prep kit, followed by post-PCR purification with AmpPure XP beads. Size selection was performed using the Blue Pippin system, and the average base pair size was determined using the Agilent TapeStation automated electrophoresis instrument. Sequencing was performed on the Illumina MiSeq platform with the 600 v3 kit. The National Microbiology Laboratory, Winnipeg, Genomics Core, conducted all sequencing and library preparation procedures. The whole genome sequences can be found on the NCBI website (bio project PRJNA1285942 and PRJNA1290672) and Table A1.

Whole genome sequence raw data were assembled [27] and annotated [28] using the Bacterial and Viral Bioinformatics Resource Center (BV-BRC v3.54.6) platform SPAdes v4.2.0 and RAST/SEED systems (https://rast.nmpdr.org/rast.cgi), respectively, using the Illumina platform with a unicycler v0.5.1. In total, all 15 SE and 24 SK whole genomes showed good quality (completeness at 100%, contamination at 0%) [29].

Summary of sequencing reads, assembly metrics, and genome Features for isolates included in This Study can be found in Appendix A.

#### 2.2.2. Phylogenetic Tree and Heatmapping

A phylogenetic tree was generated for both SE and SK isolates together using 500 single-copy gene parameters in the mafft alignment program and RAxML Fast Bootstrapping [30]. The genes in the phylogenetic tree were selected randomly from a defined number of the BV-BRC Global Protein Families to build an alignment and develop a phylogenetic tree based on the sequence differences. The number of genes ranges from 10 to 1000 for selections based on the genomes of interest [31]. A comparative system [29] was used to compare up to 500 genes and protein families using the whole genome sequences of all SE and SK isolates. Using the protein family sorter, genes of interest were searched, filtered, and grouped to examine the distribution of protein families across the genomes. A pan-genome of 5396 genes was gathered under the comparative system tool with a focus on the intra-genus comparison (PLfam) to examine the presence and absence of 124 genes and proteins of interest which play essential roles in biofilm formation, curli and cellulose production, flagella, fimbriae and pili expression, and type IV secretion system since these components were reported to be associated with biofilm development. Since all the isolates belong to the same genus (*Salmonella*), the PLfam uses only proteins within a genus, and more stringent criteria [32]. More than 100 genes were filtered and selected as they were genes associated with biofilm formation, the production of curli and cellulose, the functions of flagella, fimbriae, and pili [33]. Filtered genes were examined for their functions and were illustrated with the iTol v7 service [34].

#### 2.2.3. SNP Variation Analysis

A single nucleotide polymorphism (SNP) analysis was performed using BWA-MEM for alignment and FreeBayes as the SNP caller to identify genomic variations at specific nucleotide positions where a single base pair differs among individuals or populations [35]. SNP impacts were categorized using SnpEff, a variant annotation tool integrated into the BV-BRC pipeline. SnpEff predicts the functional consequences of each SNP based on its genomic context and classifies them into four categories, high, moderate, low, and modifier, according to the expected effect on gene or protein function. This analysis provided a comprehensive overview of potential mutations across samples. Since almost all essential protein-coding genes were present regardless of biofilm phenotype, it was vital to examine functional mutations, particularly in coding regions and upstream/downstream regulatory elements, as genes may be present but only partially functional. SNPs were examined among all studied 39 SE and SK isolates studied, using the SE isolate 53932 and SK isolate PW9-3 as the reference genomes due to their strong biofilm-forming abilities.

#### 2.2.4. BLASTp (Basic Local Alignment Search Tool for Protein) Comparison

A BLASTp comparison was used in this study to identify similarities between protein sequences and to find the best unique/unidirectional/bidirectional hit between the two trains used in the current study [36]. This sequence-based method, part of the RAST annotation pipeline [37], helped determine whether specific proteins found in strong biofilm-forming isolates were missing in intermediate, weak, or non-biofilm-forming isolates. Specifically, for the *Salmonella* Enteritidis (SE) isolates, a strong biofilm-forming poultry isolate (isolate 53932) was compared against the intermediate biofilm-forming SE isolate 107.

#### 2.2.5. BLASTp (Basic Local Alignment Search Tool for Protein) Comparison

Associations between phenotypic traits, including biofilm formation category (strong, moderate, weak, and non-producer), and the production of curli and cellulose, and genomic characteristics (SNP type and predicted SNP impact) were analyzed using the Chi-square test. Specifically, the analysis evaluated whether the distribution of isolates across biofilm formation categories was independent of genomic traits, such as the presence or absence of specific SNPs or their functional impact. Statistical significance was determined at *p* < 0.0001.

## 3. Results

### 3.1. Biofilm Formation

In this study, 15 SE and 24 SK isolates were tested for biofilm formation using the crystal violet 96-well microplate method (Section A.2) at 20–22 °C, a temperature range that reflects conditions commonly found in Canadian poultry barns, typically ranging from 19 to 25 °C [38]. Within the 15 SE isolates, results showed that 11 isolates (73.3%) produced strong biofilms, while 4 isolates (26.7%) produced intermediate biofilms (*p* < 0.0001) (Figure 1). The three intermediate biofilm producers were isolated from chicken nuggets (*n* = 2) and chicken feed (*n* = 1), indicating that SE isolated from various poultry-related sources can form biofilms at 22 °C under typical egg-layer barn temperatures. A bovine isolate also yielded an intermediate biofilm under the same conditions. Among the 24 SK isolates tested at 22 °C, only four isolates (16.7%) were strong biofilm formers, while five isolates (20.8%) were weak, and fifteen isolates (62.5%) were non-biofilm formers (*p* < 0.0001) (Figure 2).

### 3.2. Curli and Cellulose Phenotypic Results for SE and SK Isolates

Curli fimbriae and cellulose are two components typically associated with biofilm produced by *Salmonella* isolates [39]. We analyzed the production of curli and cellulose in 15 *Salmonella* Enteritidis isolates at 20 °C. Results showed that 14/15 SE isolates (93.3%) were positive for both curli and cellulose (Section A.3), with one exception, isolate #107. Interestingly, SE #107 was categorized as having intermediate biofilm formation, which could be attributed to the lack of curli and cellulose production. Conversely, not all isolates that produce curli and cellulose formed strong biofilm. Three intermediate SE isolates have had positive results for curli and cellulose. For the 24 SK isolates, 4/24 (16.7%) showed RDAR morphology for curli production and fluorescence for cellulose expression, corresponding to only four strong biofilm-forming isolates. The remaining 20 SK isolates exhibited a distinct morphology for curli expression, characterized by smooth, red, and dry colonies (SRAD) [40,41].

### 3.3. Phylogenetic Tree of the SE and SK Isolates

The phylogenetic tree, based on 500 randomly selected core genes (Figure 3), was constructed to assess the genetic relationships among SE and SK isolates with varying biofilm-forming abilities and different phenotypic results. The tree revealed two distinct major clades corresponding to SE and SK serovars, confirming their genetic divergence (branch length = 3.7 × 10^−5^), with greater similarity observed within each serovar than between them.

Within the SE isolates, two major clades emerged, referred to as the red and pink clades (Figure 3). Isolate 107 (a bovine isolate) did not cluster with either clade, forming a distinct outgroup, consistent with its bovine origin. The pink clade consisted exclusively of strong biofilm-formers, positive for curli and cellulose, whereas the red clades contained a mix, with only half of the isolates (*n* = 3) showing strong biofilm and positive curli and cellulose results. Among the 14 poultry-associated SE isolates, three of the four intermediate biofilm-formers clustered within the same red clade, suggesting some genetic basis for biofilm variation. In contrast, SK isolates formed two tightly grouped clades (green, blue + orange), with minimal genetic divergence (branch length 1.0 × 10^−6^) between members of each clade, suggesting possible clonal relationships in some of the isolates.

### 3.4. Genomic Comparison Between Different Biofilm Formation Categories and Between Serotypes

#### 3.4.1. Comparative Genomic Analysis Results with SE and SK Isolates

Using the BV-BRC Comparative Systems service and heatmap visualization, a pan-genome consisting of 5396 protein-coding genes was constructed. Among these, 4043 genes were classified as part of the core genome, shared across all analyzed isolates, while 1353 genes were designated as accessory, present in a subset of the genomes. A total of 122 protein-coding genes associated with biofilm formation, including those related to curli, cellulose, fimbriae, pili, flagella, and type IV secretion systems, were examined in these genomes, as listed in Table 2. In addition, by filtering protein-coded genes that were in the strong SE and SK isolates but absent in the other isolates, a total of 67 protein-coding genes were generated (Figure 4). Of those, 12 (CspF/H, DNA topoisomerase III, Modification methylase NgoFVII, mRNA interferase RelE, Plasmid encoded restriction endonuclease Per, Resolvase, VirB1, VirB4, VirB10, VirB6, VirB11, VirD4) had specific functions, while the remaining 55 were hypothetical proteins.

#### 3.4.2. SNP Variation Analysis

The variation analysis provides a comprehensive report of the potential mutations of the samples using SNPs. Since almost all the essential protein-coding genes linked to biofilm formation were present in nearly all samples regardless of their biofilm phenotypes, it is important to check their genomic variation, mutations, and the products of their upstream and downstream regulatory regions, since genes could be present but non-functional or partially functional.

Types of SNP variants may be found in regulatory regions (RNA-polymerase and ribosomal binding regions), coding regions, and intergenic regions (regions between the genes). Non-coding DNA is DNA that does not encode protein sequences. Some non-coding DNA can be transcribed into RNA, and even though they do not code for proteins, they are still responsible for some regulations in gene expression. Coding region SNPs consist of synonymous and non-synonymous substitutions. Synonymous substitutions change one nucleotide in a codon while still having the same amino acids after translation, and the mutations are usually silent. Nonsynonymous substitutions include missense and nonsense. A missense mutation involves changing a single base, which in turn changes the amino acids, possibly causing a malfunctioning protein. A nonsense mutation is a mutation that results in a stop codon, causing a premature stop in the translation of a protein, which could lead to a nonfunctional protein product.

#### 3.4.3. SNP Variation Analysis of 15 SE Isolates

The SNP variation analysis report (Table 3) showed a comparison of the 14 multiple-read samples of our *Salmonella* Enteritidis genomes and a closely related reference genome. SE 53932 was selected to be the reference genome due to its strong biofilm-forming ability. The report was categorized into four different SNP impacts (high, moderate, low, and modifier) and 12 specific SNP types based on the computer program SnpEff used in the BV-BRC for SNP variation analysis [42].

The results have generated a total of 62 high-impact variants, 417 moderate-impact variants, 243 low-impact variants, and 179 modifier-impact variants in all 14 SE isolates. A high-impact variant is described as having a disruptive impact on the protein, probably causing protein truncation and loss of function. In addition, the high impact could result in one or more of the following consequences: start lost, frameshift variant, missense variant, stop gained, and splice region variant.

The functions of the genes, upstream and downstream features, were found to be associated with biofilm formation development, or the major components needed for biofilm formation, such as cellulose. The association between biofilm category (strong, intermediate, weak, and non-biofilm producers) and the production of curli and cellulose was analyzed using a chi-square test, which showed a highly significant relationship across *Salmonella* Enteritidis isolates (*p* < 0.0001). However, the biofilm categories were not statistically significant or strongly associated with the different SNP types (*p* > 0.0001), whereas the phenotypic results in the production of curli and cellulose were strongly associated with the different SNP types (*p* < 0.0001). Neither the biofilm categories nor the production of curli and cellulose were yet strongly associated with the SNPs impact types (high, moderate, low, or modifier) (*p* > 0.0001).

#### 3.4.4. SNP Variation Analysis of 24 SK Isolates

The variation analysis report showed a comparison of 23 multiple-read samples of our SK isolates genomes and a closely related reference genome. The SK isolate PW9-3 isolated from plucking water was selected to be the reference genome due to its strong biofilm-forming ability. The results have generated a total of 19 moderate variants, 53 low-impact variants, and 39 modifier impacts. From the total of 111 output of genetic variations and mutations, 43 possible genes, proteins, and enzymes were revealed. Of those 43 potential candidates, 3 of them were selected upon screening after comprehensive reviews of the literature, and these include *tolA, tolB,* and *tolR* (Table 3).

### 3.5. SE Isolates Functional Genomic Comparison (Strong vs. Intermediate Biofilm-Forming Isolates)

To examine how the intermediate bovine isolate was different from the other strong poultry isolates, a proteome comparison was conducted. A strong SE biofilm isolate 53932, as a reference, was blasted against an intermediate SE isolate 107. A graph is provided in Figure 5, and a detailed table is provided in Table 4. Out of 4765 protein-coding gene sequences that were compared against each other, 148 of them had 0% identity, indicating that a specific protein was present in the strong reference genome but absent in the intermediate comparison genome. Excluding hypothetical proteins and proteins with unknown functions, 65 proteins remained. Within those 65 proteins, a group of T4SS proteins was especially associated with biofilm formation and virulence factors. VirB1, VirB3, VirB4, VirB5, VirB6, VirB9, VirB10, VirB11, VirD2 homolog, and VirD4 were present in the strong biofilm SE isolate genome but absent in the intermediate one. According to our results, VirB1, VirB3, VirB4, VirB5, VirB6, VirB10, VirB11, and VirD4 were present in 4 strong SE isolates, and absent in the rest of the SE isolates and all SK isolates. The detailed mechanisms underlying biofilm formation and gene expression involve multiple regulatory pathways that are complex and need more investigation. As a result, the broader implication of these findings remains elusive.

A group of six SE isolates (three strong isolates: 52239, 51094, and 51095 (9susph), and three intermediate isolates 715701, 79901, and 79801), which belonged to the same clade in red color in Figure 3 were compared by their functional genomic information due to their clustered SNPs results (Figure 6). One of the strong biofilm-forming isolates 52239, served as the reference isolate. Comparing the strong reference isolate 52239 to the intermediate isolate 79901, excluding the hypothetical proteins and proteins with unknown functions, only two proteins were absent in 79901 that were present in 52239: secreted effector protein and Thiol: disulfide interchange protein DsbA precursor, which indicated that this intermediate isolate was not too different from the strong isolate. However, when comparing 715701 and 79801 to the reference isolate, many proteins appeared to be absent. Table 5 revealed the proteins present in the reference isolate 52239, but lacking in either the intermediate isolate 715701 (^a^) or 79801 (^b^).

## 4. Discussion

### 4.1. Biofilm Formation

Among the 15 *Salmonella* Enteritidis (SE) isolates, 11 (73.3%) produced strong biofilms, while 4 (26.7%) produced intermediate biofilms (*p* < 0.0001). Biofilm results were in alignment with the curli and cellulose results, which were 93% positive for both traits. These findings clearly demonstrate that SE isolates from poultry sources can form strong biofilms at 20–22 °C, a trait that may contribute to the persistence of SE strains in the environment. Studies from Brazil present contrasting results. De Oliveira et al. found that SE isolates incubated for 96 h, at 20 °C on various surfaces (stainless steel, glass, and PVC) were predominantly non-biofilm formers [43]. Similarly, Rodrigues et al. reported that among 171 SE isolates from foodborne outbreaks (foods and stool samples) and poultry products, most were weak or non-biofilm formers after 24 h at 25 °C using the crystal violet method at 550 nm in the biofilm reader [19]. Both studies highlight that isolates with strong biofilm-forming capacities were a minority. These results suggest that SE isolates from Brazil possess different biofilm formation abilities compared to those isolated in North America, possibly associated with distinct genomic traits or methodological differences, such as incubation times (24/48/120 h) and broth compositions (TSB vs. LB-NS).

Regarding *Salmonella* Kentucky (SK), of the 24 isolates tested at 22 °C, only four (16.7%) were strong biofilm formers, five (20.8%) were weak formers, and the majority, fifteen (62.5%), were non-biofilm formers (*p* < 0.0001). Notably, among the four strong biofilm formers, only curli and cellulose production were positive for both traits, which may help explain the overall lower biofilm-forming capacity observed in this group. These results demonstrate that the biofilm-forming ability of SK isolates was markedly lower compared to SE; perhaps this could limit SK’s ability to persist in poultry barn environments. SK results on biofilm formation are similar to those reported by Obe et al., where biofilm formation of SK (*n* = 12) isolated from commercial processing equipment ranged in biofilm production from strong (*n* = 3), to moderate (*n* = 2), to weak biofilm producers (*n* = 7) after incubation in TSB for 48 h at 25 °C [44]. In our study, SK isolates from similar processing environments exhibited a wide range of biofilm abilities under the specific conditions tested. For example, of the eight SK isolates isolated from the carcass wash/pre-chill step, seven were non-biofilm formers, while one was a strong biofilm former. Similarly, of the ten SK isolates isolated from the final wash/post chill, nine were non-biofilm formers and one was a strong biofilm former. Biofilm formation is also strongly influenced by incubation time and nutrient availability. Stepanović et al. demonstrated that biofilm production peaked at different times depending on temperatures, with stronger biofilm formation at 22 °C after 48 h [45]. Piras et al. found similar temperature-dependent variability in *Salmonella* isolates from pig slaughterhouses [46]. Ziech et al. also reported that most *Salmonella* spp. isolated from poultry cutting rooms were weak or intermediate adherents after 96 h at 35 °C [47]. Another key factor affecting biofilm formation is the type of growth medium. Our study utilized Lennox Broth No-Salt (LB-NS), whereas most other studies employed tryptic soy broth (TSB), which may yield different results in biofilm formation.

The strong biofilm formation observed in SE and some SK isolates suggests that these strains have an enhanced capacity for persistence. Thus, thorough cleaning and sanitation in poultry environments and other processing environments are critical. Effective scrubbing, correct use of detergents, and ensuring proper sanitizer concentration and contact time are essential to prevent surface conditioning, bacterial attachment, and subsequent biofilm development. SE and SK results on biofilm formation abilities raises a critical question: can genomic features be used to differentiate *Salmonella* strains that are capable of form biofilms and persist in the environment from those that are more easily eliminated. Genomic analysis may help address this gap by identifying genetic determinants associated with biofilm formation, thereby providing a clearer picture of the factors driving strain survival.

### 4.2. Genomic Analysis

To further explore the genetic relationships among the isolates, a phylogenetic tree was constructed including SE and SK (Figure 3). In the phylogenetic tree, the absence of clustering by source indicates that isolates from different geographic locations are not confined to specific environments. Instead, they are genetically intermixed, which suggests that these *Salmonella* strains possess broad fitness traits rather than niche adaptations. This pattern reflects a high degree of adaptability, allowing them to persist and survive under the diverse and often harsh conditions encountered throughout poultry production and processing environments [48]. The bootstrap values, represented as percentages in the tree, indicate the frequency with which the same branch is observed when the phylogenetic tree is generated 100 times [49]. A bootstrap value of 100% indicates that this arrangement was observed every time, confirming that the result is reliable and not due to random chance, while 0% indicates a random ordering of the isolates within clades [49]. Within the SE isolates, three groupings were confidently supported: 79801 and 71501 (100%), 619102 and 618901 (91%), 53932, 53936, and 61401 (98%), while SK isolates consistently grouped with 100% support, though individual branch supports were low (0%), further suggesting low divergence. Horizontal branch lines in the tree represent the number of genetic changes and evolutionary lineages change over time, with the branch length representing the number of substitutions per 100 nucleotide sites [50]. The shorter the horizontal lines, the fewer genetic base differences along the sequences, suggesting closer relatedness between the isolates. Indeed, all of the SK isolates in blue and yellow diverge by only 1 base substitution per million bases, using the 500 selected genes in the analysis. The phylogenetic analysis offered limited resolution of phenotypic and genomic differences among isolates. While it supports the hypothesis that genetic variation underlies differences in biofilm-forming capacity in SE isolates, no clear clustering aligned with phenotypic traits, underscoring the need to examine functional mutations in core genes and the regulatory region. Although *Salmonella* Kentucky serovar is said to be polyphyletic globally [51], all isolates in our study clustered within a single well-supported clade, indicating a monophyletic origin. This suggests a shared ancestry and possibly a common source or limited diversity among these poultry-associated isolates.

The comparative genomic analysis showed that all 39 *Salmonella* isolates (SE and SK) carried nearly all genes previously reported to encode proteins involved in biofilm formation, including those related to curli and cellulose production, flagella, fimbriae, pili, and type IV secretion systems, regardless of their phenotypic biofilm formation category, curli/cellulose production, or isolation source. The presence or absence of these genes did not correlate with the observed biofilm formation phenotypes. Several key genes associated with biofilm development and regulation, such as *bap*A, *crl*, and *ddh*C, were absent in all 15 SE and 24 SK isolates. In contrast, some proteins, including YadK, YhcA, and EcpD, were lacking in SE isolates but present in SK isolates. Notably, genes *csg*DEF were missing in the non-biofilm-forming SK isolate F41-2, and type IV secretion system proteins (VirB1, VirB3, VirB4, VirB6, VirB10, VirB11, and VirD4) were absent from all 24 SK isolates but present in four strong biofilm-forming SE isolates. Interestingly, isolate #107, the only bovine isolate and the only isolate with negative results for both curli and cellulose production, carries all the screened genes except those type IV secretion system genes (Figure 4).

All 39 isolates carried four protein-coding genes linked to biofilm regulation: *bssS*, *bssR*, *yjgK*, and the *bdcA (cyclic*-di-GMP-binding biofilm dispersal mediator protein) (Figure 4). The genes *bssS* and *bssR* act as negative regulators of biofilm formation, reducing biofilm mass, surface coverage, and mean thickness in *Salmonella* and *E. coli* [52,53]. It has been reported that overexpression of *yjgK* has been shown to repress fimbrial genes, reducing biofilm formation at 8 h but enhancing it after 24 h. The *bdcA* gene facilitates biofilm dispersal using c-di-GMP [54]. Cyclic-di-GMP further illustrates the complexity of biofilm regulation, acting antagonistically by suppressing motility, dispersal, and virulence gene expression in single-cell states, while promoting sessility, adhesion, and biofilm development in multicellular states [55]. Interestingly, *bapA*, which encodes a cell-surface protein critical for biofilm formation, was absent in all 39 isolates. Previous studies have shown that deletion of *bapA* abolishes biofilm formation, while overexpression enhances it. The absence of this gene in our isolates suggests that alternative pathways or compensatory mechanisms are driving biofilm development, highlighting the genetic diversity and regulatory complexity underlying this phenotype in *Salmonella.*

Regarding the *csg* (curli specific gene) family, essential for curli and cellulose production [33,56], it was found that genes *csgA*-*csgG* were present in all 15 SE isolates and 23 out of 24 SK isolates. In contrast, genes *csgDEF* were absent in the SK F41-2 isolate, a non-biofilm former. Two key operons regulate curli production: *csgBAC* and *csgDEFG*. Among them, *csgA*, *csgB*, and *csgD* are particularly crucial. Gene *csgA* directly contributes to curli fiber formation, a key factor in biofilm adhesion and stability [51], while *csg*B stabilizes *csg*A expression and promotes the initial aggregation of *csgA* to form curli fibers [57]. Gene *csgC* and genes *csgEFG* function as accessory genes involved in *csgA* polymerization [58] and in the assembly, secretion and structural integrity of curli fibers [59]. The absence of *csgD* in SK F41-2 explains its non-biofilm-forming phenotype. Several proteins that regulate *csgD* expression were found in our *Salmonella* genomes: RpoS, OmpR, Crl, and MlrA. It has been reported that the transcription of *csgD* was utterly dependent on the expression of RpoS and OmpR proteins [60,61,62], both of which were present in all 39 *Salmonella* isolates. However, Crl, an RopS-binding factor, was absent in all 39 isolates. Crl protein interacts with RpoS and it is required for the maximal expression of the *csgD*, *csgB*, *adrA*, and *bcsA* genes, which are essential for the curli and cellulose synthesis [63]. MlrA, present in all strains, is a novel transcriptional regulator of *csgD* that acts as a positive regulator of RpoS-dependent curli production in *Salmonella* Typhi and *E. coli* [64]. These findings suggest that *csg*D regulation in these strains may rely exclusively on RpoS and OmpR, without the modulatory influence of Crl. Interestingly, while all non-biofilm formers (except SK F41-2) harbored all seven *csg* genes, only 4 out of 24 SK isolates exhibited the red, dry, and rough (RDAR) curli morphotype [60]. Gene *ddhc* (formerly *rfbH*), which codes for CDP-4-keto-6-deoxy-D-glucose-3-dehydrase, is responsible for synthesizing abequose, the last sugar component of the O antigen in LPS. Gene *waaG* (formerly *rfaG*), encodes UDP-glucose-LPS α1,3-glucosyltransferase, an enzyme that adds a glucose molecule to heptose II of the core polysaccharide of LPS [40]. Both mutations result in decreased levels of curli while increasing cellulose production. The gene *waaG* was present in all 15 SE and 24 SK isolates, while the gene *ddhC* was absent in all isolates. Although all non-biofilm formers (except SK F41-2) harbored the complete set of *csg* genes (*csgA*–*csgG*), most of them did not exhibit the RDAR morphotype. This disconnect between gene presence and phenotype demonstrates that biofilm formation is not determined solely by the presence of the *csg* gene cluster, but instead depends on regulatory dynamics and additional factors.

All isolates carried *bcsABCEFGQ*, with three intermediate SE isolates showing a duplicated *bcsC*, which may be linked to reduced biofilm formation. In contrast, *bcsZ* and *yhj*R were absent in all isolates. The diguanylate cyclase gene *adrA*, a positive regulator of *csgD*-dependent cellulose synthesis, was present in all strains. SNP analysis revealed mutations in several *bcs* genes, suggesting possible functional alterations that could affect cellulose production; these will be discussed in later sections. Overall, while cellulose biosynthetic operons are broadly conserved, genetic variations such as *bcsC* duplication, absence of accessory genes, and SNPs in key loci may influence the balance between curli and cellulose production, contributing to observed biofilm phenotypes.

We also looked at fimbriae and pili genes. Of the 35 fimbriae protein-coding genes analyzed, 32 of them were present in all *Salmonella* isolates. However, YadK, YhcA, and EcpD were absent in all SE isolates but present in all SK isolates (Figure 4). The adhesion operon Yad has several genes, including *yadK*. In *E. coli*, the fimbria Yad contributes to biofilm formation and bladder epithelial cell adherence [65]. YhcA is an uncharacterized fimbrial chaperone protein, while the *ecp* operon encodes *ecpD*, which is necessary for pili assembly and functions as a polymerized tip adhesin [66]. The absence of these three genes in SE isolates remains unexplained. YehA protein was found in both serovars, but had different amino acid lengths and was present in two copies in 6 SK isolates. Additionally, a total of 8 Type IV pilin genes were present in all isolates, including *pilABCMNOPQ*, which encode bacterial appendages involved in adhesion and motility.

The flagellar filament and its motility in *Salmonella* play an important role in the early development of biofilms, which is the initial attachment to the surface, rather than contributing to the biofilm maturation [14]. In our study, a total of 45 flagellar-related protein-coding genes were found, of which, 18 belong to the Fli family (FliC-FliT), 14 belong to the Fla family (FlgA-FlgN), 5 Flh proteins (FlhA-FlhE), motor rotation proteins (MotA and MotB), flagellar regulator proteins (flk, RtsA, and RtsB), chemotaxis regulator, and RNA polymerase sigma factor for flagellar operon. All of which were present in all isolates (Figure 4). In the SNP variation analysis, several proteins had mutations in the SE isolates, including MotA, MotB, FlgA, FlgB, FlgM, FlhC, FliT, and RtsB, which will be discussed in later sections.

Virulence factors such as the type III secretion system (T3SS) have been studied extensively regarding biofilm formation in *Salmonella* isolates. However, all of the protein-coding genes associated with T3SS were present in all isolates in our study. Interestingly, type IV secretion system (T4SS) proteins have shown different results. T4SS plays an important role in mediating the translocation of macromolecules across bacterial cell envelopes [67]. Out of 8 proteins found, 1 of them (NAD(P)H dehydrogenase) was present in all isolates. Seven of them were only found in 4 strong-biofilm SE isolates but were absent in the rest of the 11 SE and all 24 SK isolates. Those proteins include peptidoglycan hydrolase (VirB1), inner membrane protein (VirB3, VirB6, and VirB10), ATPase (VirB4 and VirB11), and coupling protein VirD4. This result suggests that these groups of T4SS proteins may influence biofilm formation or its related mechanisms. Similarly to the *Agrobacterium tumefaciens* VirB/D T4S system, the T4S system in Gram-negative bacteria typically consists of 12 proteins: VirB1-11 and VirD [67]. Of those 12 core proteins, 7 of them (VirB1, VirB3, VirB4, VirB6, VirB10, VirB11, and VirD4) were detected in 4 of the 15 SE isolates, but none were detected in the SK isolates (Figure 4). To date, only a few *Salmonella* serotypes, such as *Salmonella* Enteritidis phage type 34, have been reported to carry T4SS [68]. The functions of T4SS genes and their relationship with biofilm formation are still elusive and need further investigation.

For the SNPs variation analysis of the SE isolates, three curli genes were found to have a moderate-impact nonsynonymous missense mutation in the curli gene *csgF*, its upstream feature *csgG*, and its downstream feature *csgE* in SE 52239 (a strong biofilm-former), changing the codon TCA into CCA. Gene *csgEFG* are all components for curli production assembly and transportation. These are considered accessory genes, which might explain why SE 52239 remained a strong biofilm producer despite the presence of a missense mutation.

This research found genetic mutations in SE isolates in essential proteins responsible for cellulose and curli biosynthesis (Table 3). As discussed in the cellulose section, *bcs* operon *yhjR-bcsQABZC* encodes genes *yhjR*, *bcsQ*, *bcsA*, *bcsB*, *bcsZ*, and *bcsC*, and operon *bcsEFG* encodes gene *bcsE*, *bcsF*, and *bcsG*. Out of 5 mutations associated with cellulose biosynthesis, one was stop-gained, one was frameshift deletion mutation, and three were synonymous variants. Gene *bcsA*, its upstream feature *bcsQ*, and its downstream feature *bcsB* had a high-impact stop-gained nonsynonymous mutation with a codon changed from TGG to TAG. Stop codons signal the end of the translation, when it occurs, it can result in a truncated (shortened) protein and disrupt the normal function of this protein. The stop-gained mutation was found in 6 SE isolates (intermediate formers: 715701, 79801, 79901, strong formers: 52239, 51094, and 51095). This group corresponded to the red clade in Figure 3. Having both intermediate and strong isolates present indicates that this mutation might not be associated with loss or reduction in biofilm functions, or that only three intermediate isolates were impacted by this mutation. Gene *bcsC* had a high-impact frameshift deletion mutation in the 3 intermediate isolates mentioned above and a low-impact synonymous variation in isolate 107. Frameshift deletion mutation is missing nucleotides “cg”. Interestingly, *bcs* mutations were not seen in SK isolates, where non-biofilm formers were predominant. These *bcs* genes are essential in synthesizing cellulose in the matrix. The fact that all three isolates associated with this frameshift deletion mutation were intermediate biofilm formers indicates that this mutation might be associated with a reduction in the biofilm functions. Gene *bcsG* and its downstream feature *bcsF* had a low-impact synonymous variation in isolate 107, indicating that isolate 107, being synonymous, was most likely not associated with a reduction in biofilm formation. However, it is worth noticing that isolate 107 is a quite distant relative of the other SE isolates. Gene *bcsQ* and its upstream feature YhjR also had a low-impact synonymous variation in isolate 107. The deletion of gene *bcsQ* and *yhjR* could result in no or reduced cellulose production in *E. coli* [69]. However, synonymous variation should not alter the amino acid sequence.

Yeh, Yad, and Yfc are all core-associated chaperon-usher fimbriae-related proteins that are conserved in *E. coli* and *Salmonella* isolates [70]. A high-impact frameshift insertion in *yeh*A and its upstream feature *yeh*B was detected in six isolates (three intermediate and three strong biofilm formers), suggesting the mutation is not directly linked to reduced biofilm capacity. In isolate 107, *yad*E and its upstream feature *yadI* carried synonymous variants with no expected functional effect, while *yad*S harbored a moderate-impact missense mutation that may contribute to reduced biofilm formation. Similarly, *yad*U showed moderate-impact missense variants in isolate 107 and three intermediate isolates, suggesting a possible role in biofilm modulation. In contrast, synonymous variants in *yfc*L and *yfc*C in isolate 107 are unlikely to affect protein function.

Flagella are another critical component in biofilm formation. Several flagellar-associated genes were detected with genetic variations and mutations. Gene *flgA*, its upstream feature, *flgM*, and its downstream feature, *flgB*, had moderate-impact missense mutations in all six isolates mentioned above. *motA*, its upstream feature *motB*, and its downstream feature *flhC* had low-impact synonymous variations in isolate 107. *fliT* had a missense mutation and *rtsB* had a synonymous variation, both in isolate 107.

For the SNP variation analysis of SK isolates, gene *tolA* and *tolR* are both part of the Tol-Pal system of the *E. coli* envelope, which consists of *tolQ*, *tolR*, *tolA*, *tolB*, and *pal* [71] and this system is involved in maintaining the integrity of the outer membrane [72]. More importantly, *tolA* can promote survival, biofilm formation, and virulence of avian pathogenic *E. coli* [73]. According to Su et al., a *tolA* mutation with a motility defect results in significantly less biofilm biomass, a reduced amount of *fliR* expression, and weakened resistance to environmental The regulation of the *fnr* gene controls the expression of TolA [74]. Therefore, even though *tolA*, *tolB*, and *tolR* genes are present in all isolates with 1 copy in each isolate except for *tolA* gene in SK F43-3 and SK W39-1 (non-biofilm producers), which have two copies, it is possible that the changes in the amino acids in the nonsynonymous mutation (1 out of 5 *tolA* mutations found in SK isolates, 3 out of 4 in SE isolates) missense variations altered the protein’s structure and function, and lead to reduced biofilm formation and the flagellar biosynthesis (Table 3). Both serovars had synonymous and nonsynonymous mutations. This could explain the strong biofilm formation in SK B38-3, since it did not have this mutation, and intermediate biofilm formation in SE 715701 and SE 79801, since they have this mutation. In summary, the three SE intermediate isolates exhibited more high-impact variations than the others, and none of the SK stains showed any high-impact variation. These findings suggest that while both serovars carry *tolA*, *tolB*, and *tolR*, nonsynonymous mutations, especially in *tolA*, may compromise protein function and reduce biofilm capacity, with SE isolates showing more high-impact variations than SK. This highlights *tolA* as a potential key determinant of biofilm formation differences between isolates.

## 5. Conclusions

The results revealed distinct genetic patterns differentiating strong from weak and non-biofilm-producing isolates. Although most isolates carried the core genes for biofilm formation (curli, cellulose, fimbriae, pili, and flagella), the SK isolates were predominantly non-biofilm producers despite possessing nearly all essential genes. No major SNPs were detected in core biofilm genes, except mutations in *bcs* (isolate#107) and *csg* (strong SE 52239). Notably, SE isolates showed the highest variation, with three high-impact mutations: a nonsynonymous stop-gain in *bcsA* (six isolates: three intermediate, three strong), a frameshift deletion in *bcsC* (three intermediate), and a frameshift deletion in *yehA* (the same six SE isolates). These variations suggest that SE, more than SK, may rely on genetic changes to modulate biofilm capacity.

Furthermore, strong SE isolates carried a cluster of type IV secretion system proteins (VirB1, VirB3, VirB4, VirB5, VirB6, VirB9, VirB11, VirD2 homolog, and VirD4) on a lysogenic phage, absent in intermediate strains. Additional biofilm-associated genes were linked to a conjugative plasmid. Together, these findings imply that beyond SNPs in core genes, horizontal gene transfer through phages and plasmids may play a crucial role in enhancing SE biofilm formation. This highlights the need for future studies to clarify how genetic variations and mobile elements interact to drive biofilm development.

Variations in *tol*A, *tol*B, and *tol*R genes, particularly nonsynonymous mutations in *tol*A, may influence protein function and reduce biofilm-forming capacity, with *Salmonella* Enteritidis isolates showing more high-impact mutations than S. Kentucky. These findings identify *tol*A as a potential determinant of biofilm variability; however, further functional studies are needed to verify its specific role in regulating biofilm formation across different *S*. Enteritidis isolates.

A key limitation of this study is that the isolates of *Salmonella* Enteritidis (SE) and *Salmonella* Kentucky (SK) were not selected to capture the full genetic diversity within each serovar. It is possible that the isolates tested were clonally related, which could explain the similarity in biofilm formation and underlying genetic traits observed within each group. As a result, the findings may not fully represent the broader variability in biofilm-forming capacity across genetically diverse isolates of SE and SK. Future studies should therefore include a wider range of isolates to better account for intra-serovar diversity and provide more generalizable insights.

## Figures and Tables

**Figure 1 microorganisms-13-02473-f001:**
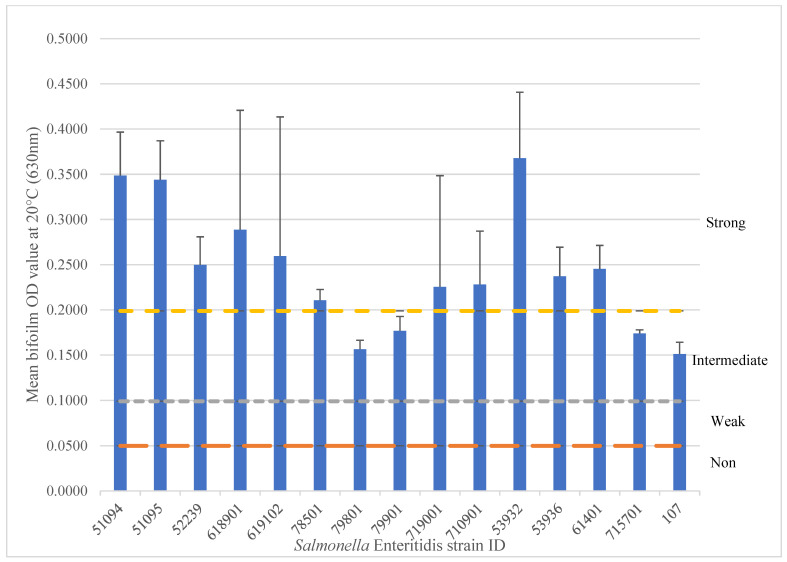
*Salmonella* Enteritidis isolates form biofilms at 22 °C after 5 days. The biofilm formation ability is categorized by strong (above the yellow line), intermediate (above the gray line), weak (above the orange line), and non-biofilm formers (below the orange line). The experiment was repeated three times.

**Figure 2 microorganisms-13-02473-f002:**
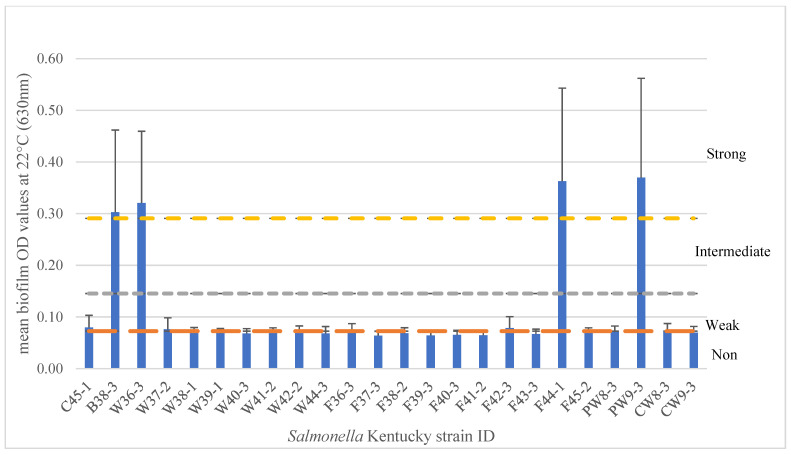
*Salmonella* Kentucky isolates biofilm formation at 22 °C after 5 days. The biofilm formation ability is categorized by strong (above the yellow line), intermediate (above the gray line), weak (above the orange line), and non-biofilm formers (below the orange line). The experiment was repeated four times.

**Figure 3 microorganisms-13-02473-f003:**
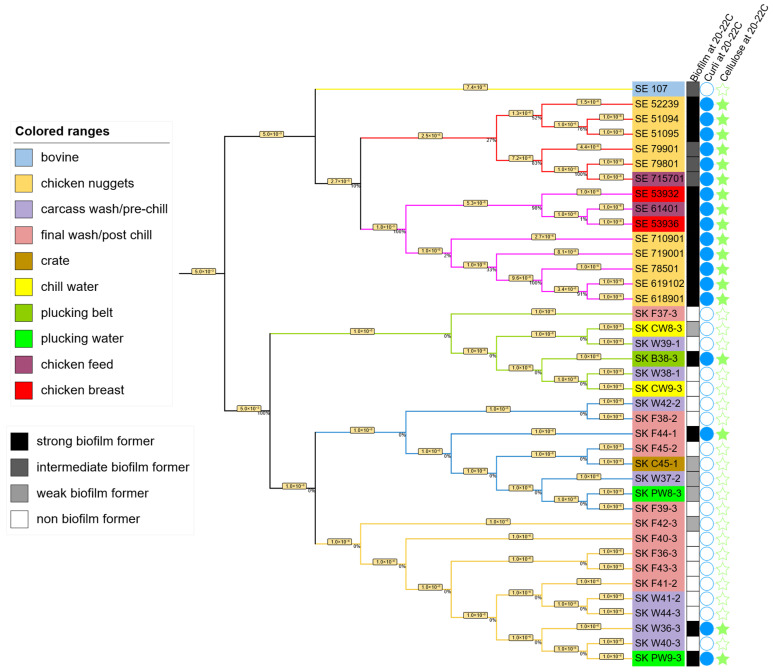
Phylogenetic tree of 15 SE isolates and 24 SK isolates using 500 single-copy genes parameter with the mafft alignment program and RAxML Fast Bootstrapping branch supporting method. The tree was divided into two big clades. The upper part belongs to the 15 SE isolates (further divided into the red clade, pink clade, and outgroup isolate #107), and the lower part belongs to the 24 SK isolates (further divided into the green, blue, and orange clades). The numbers in percentages indicate the level of confidence in this clustering method, and the number in the horizontal line represents the number of nucleotide changes or substitutions per site, indicating the closeness of those isolates. The isolates IDs were color-coded to show their sources of isolation. Filled (solid) symbols indicate positive results for biofilm formation, curli and cellulose production. The black-to-white color scale represents the biofilm formation category (strong, intermediate, or weak). Empty (hollow) symbols indicate negative results (no significant biofilm formation).

**Figure 4 microorganisms-13-02473-f004:**
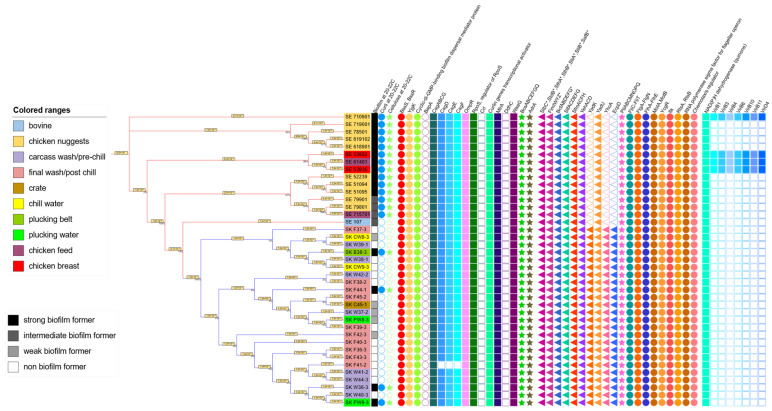
Biofilm formation results and binary data of the biofilm-related genes in all 39 isolates. For the biofilm development assay, the intensity of the shaded squares (black-white scale) represents biofilm-forming ability: black = strong, medium dark gray = intermediate, light-gray = weak, and white = non-biofilm former. For the production of curli and cellulose, a filled shape indicates a positive result and an empty shape indicates a negative result. For the genotypic results (heat map on the right side), the genes associated with biofilm formation, curli, cellulose, fimbriae, flagella, pili, and T4SS are shown with the gene names at the top of the image. Solid figures represent gene presence and empty ones, gene absence. The * in the gene name represents the 7 cluster groups in the flagella family. The left side shows a dendrogram that illustrates the genetic relatedness among all 39 Salmonella isolates. Each branch point (node) represents a common ancestor between the isolates or groups connected by that node. The branch length reflects the degree of genetic difference; shorter branches indicate closely related isolates, while longer branches indicate greater genetic divergence. Clusters of isolates that share both short branch lengths and the same color suggest high genetic similarity among isolates from the same environment or sample type. The bootstrap values displayed at the nodes represent the level of confidence in each branch grouping (higher values indicate stronger support for that relationship).

**Figure 5 microorganisms-13-02473-f005:**
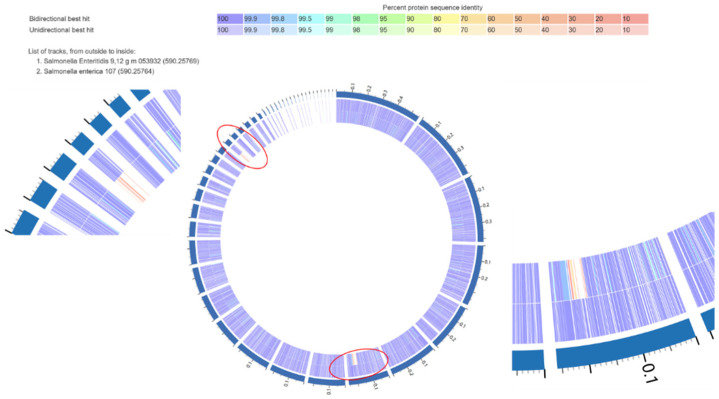
Protein coding gene comparison graph showing differences in gene sequence identity between a strong SE biofilm isolate isolated from poultry product (53932) on the outside ring, and an intermediate SE isolate isolated from bovine (107) on the inside ring. The image shows the plot of the bi-directional BLAST hits and a legend that shows the list of tracks and the strength of the BLAST hits, where purple is the strongest and red is the weakest. Two regions of the cluster of differences in the red circles were enlarged for better visualization.

**Figure 6 microorganisms-13-02473-f006:**
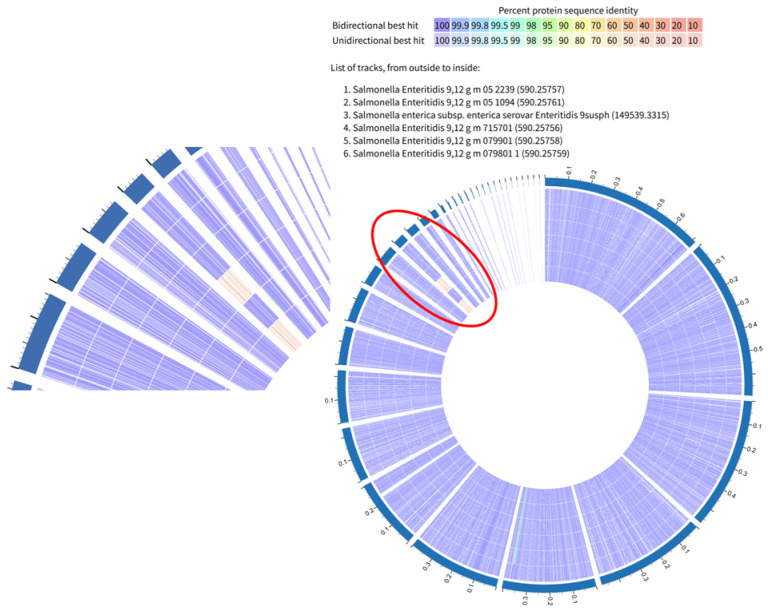
Protein coding gene comparison graph showing differences in gene sequence identity between three strong SE biofilm isolates (52239, 51094 and 51095 (another name 9susph)) on the outside ring, and three intermediate SE isolates (715701, 79901, and 79801), with 52239 being in the most outside ring and 79801 being in the most inside ring. The image shows the plot of the bi-directional BLAST hits and a legend that shows the list of tracks and the strength of the BLAST hits, where purple is the strongest and red is the weakest. One region of cluster of differences in the red circles was enlarged for better visualization. The red circle indicated an obvious difference among those isolates.

**Table 1 microorganisms-13-02473-t001:** List of *Salmonella enterica* serovars Enteritidis and Kentucky used in this study.

Serovar	Isolate ID	Source	Serovar	Isolate ID	Source
Enteritidis	51094	Chicken nuggets	Kentucky	C45-1	Crate
Enteritidis	51095	Chicken nuggets	Kentucky	B38-3	Plucking belt
Enteritidis	52239	Chicken nuggets	Kentucky	W36-3	Carcass wash/Pre-chill
Enteritidis	618901	Chicken nuggets	Kentucky	W37-2	Carcass wash/Pre-chill
Enteritidis	619102	Chicken nuggets	Kentucky	W38-1	Carcass wash/Pre-chill
Enteritidis	78501	Chicken nuggets	Kentucky	W39-1	Carcass wash/Pre-chill
Enteritidis	79801	Chicken nuggets	Kentucky	W40-3	Carcass wash/Pre-chill
Enteritidis	79901	Chicken nuggets	Kentucky	W41-2	Carcass wash/Pre-chill
Enteritidis	719001	Chicken nuggets	Kentucky	W42-2	Carcass wash/Pre-chill
Enteritidis	710901	Chicken nuggets	Kentucky	W44-3	Carcass wash/Pre-chill
Enteritidis	53932	Chicken breast	Kentucky	F36-3	Final wash/post chill
Enteritidis	53936	Chicken breast	Kentucky	F37-3	Final wash/post chill
Enteritidis	61401	Feed	Kentucky	F38-2	Final wash/post chill
Enteritidis	715701	Feed	Kentucky	F39-3	Final wash/post chill
Enteritidis	107	Bovine	Kentucky	F40-3	Final wash/post chill
			Kentucky	F41-2	Final wash/post chill
			Kentucky	F42-3	Final wash/post chill
			Kentucky	F43-3	Final wash/post chill
			Kentucky	F44-1	Final wash/post chill
			Kentucky	F45-2	Final wash/post chill
			Kentucky	PW8-3	Plucking water
			Kentucky	PW9-3	Plucking water
			Kentucky	CW8-3	Chill water
			Kentucky	CW9-3	Chill water

**Table 2 microorganisms-13-02473-t002:** Screened genes, protein groups associated with biofilm formation, curli and cellulose production, pili, flagella, and fimbriae, and their functions (functions were provided by the BV-BRC website).

Gene Category	Protein Name	Function
Biofilm regulation protein-coding genes	BdcA (Yjgl)	Cyclic-di-GMP-binding biofilm dispersal mediator protein
	BssS/BssR	Biofilm regulator
	YjgK	Linked to biofilm formation
Biofilm structural protein	BapA	A large cell-surface protein required for biofilm formation by S. Enteritidis
Curli production	CsgA	Major curlin subunit precursor
	CsgB	Minor curlin subunit, nucleation component of curlin monomers
	CsgC	Putative curli production protein
	CsgD	Transcriptional regulator for the 2nd curli operon
	CsgE/CsgF, CsgG	Curli production assembly/transport component
	OmpR	Two-component system response regulator
	RpoS	RNA polymerase sigma factor
	Regulator of RpoS	
	Crl	An RpoS-binding factor, binding to RpoS, facilitates RNA polymerase holoenzyme formation (EσS)
	Curlin gene transcriptional activator	
	MlrA	HTH-type transcriptional regulator; Positive regulator of CsgD expression
	DdhC	Involved in the synthesis of abequose
	WaaG	UDP-glucose:(heptosyl) LPS alpha1,3-glucosyltransferase
Cellulose production	BcsA	Cellulose synthase catalytic subunit
	BcsB	Cyclic di-GMP-binding protein
	BcsC	Cellulose synthase operon protein C
	BcsE/BcsF/BcsG/BcsQ	Cellulose biosynthesis protein
	AdrA (DgcC)	Diguanylate cyclase positively regulates cellulose synthesis via production of the secondary messenger signaling molecule (3′-5′)-cyclic diguanosine monophosphate (c-di-GMP)
Fimbriae production	StbC */StdB */SthB */StiC *	Fimbriae usher protein
	SthA */StiB *	Putative fimbrial chaperone
	StiA *	Putative fimbrial subunit
	SafB *	*Salmonella* atypical fimbria periplasmic chaperone
	FimI *	Fimbriae-like adhesin
	FimW *	Fimbriae W protein
	FimY */FimZ *	Transcriptional regulator of fimbriae expression
	BcfA */BcfD */BcfE */BcfF *	Fimbrial subunit
	BcfB */BcfG *	Fimbrial chaperone
	StfA	Major fimbrial subunit
	StfC	Fimbriae usher protein
	StfD	Periplasmic fimbrial chaperone
	StfE/StfF/StfG	Minor fimbrial subunit
	SfmA/SfmH/YehD/SfmF	Uncharacterized fimbrial-like protein
	SfmC/YehC	Probable fimbrial chaperone
	YehA/YadK	Uncharacterized fimbrial-like protein
	YadU	Uncharacterized protein in the stf fimbrial cluster
	YhcA	Uncharacterized fimbrial chaperone
	EcpD	Fimbria adhesin
Pili	PilA	Type IV pilin
	PilB	Type IV pilus assembly, ATPase
	PilC	Type IV pilus assembly protein
	PilM/PilN/PilO/PilP/PilQ	Type IV pilus biogenesis protein
Flagella	FliC	Flagellin
	FliD	Flagellar cap protein
	FliE	Flagellar hook-basal body complex protein
	FliF	Flagellar M-ring protein
	FliG	Flagellar motor switch protein
	FliH	Flagellar assembly protein
	FliI	Flagellum-specific ATP synthase
	FliJ	Flagellar protein
	FliK	Flagellar hook-length control protein
	FliL	Flagellar basal body-associated protein
	FliM/FliN	Flagellar motor switch protein
	FliO/FliP/FliQ/FliR/FliS/FliT	Flagellar biosynthesis protein
	FlgA	Flagellar basal-body P-ring formation protein
	FlgB/FlgC/FlgF/FlgG	Flagellar basal-body rod protein
	FlgD	Flagellar basal-body rod modification protein
	FlgE	Flagellar hook protein
	FlgH	Flagellar L-ring protein
	FlgI	Flagellar P-ring protein
	FlgJ/FlhE	Flagellar protein
	FlgK/FlgL	Flagellar hook-associated protein
	FlgM	Negative regulator of flagellin synthesis
	FlgN/FlhA/FlhB	Flagellar biosynthesis protein
	FlhC/FlhD	Flagellar transcriptional activator
	MotA/MotB	Flagellar motor rotation protein
	YcgR	Flagellar brake protein
	flk	Flagellar regulator
	RtsA	Type III secretion and flagellar regulator
	RtsB	Flagellar regulon repressor
	Chemotaxis regulator	Transmits chemoreceptor signals to flagellar motor components CheY
	RNA polymerase sigma factor for the flagellar operon	
Type IV secretion system	NAD(P)H dehydrogenase (quinone), VirB4, VirB11	ATPase is required for both the assembly of the type IV secretion complex and the secretion of the T-DNA complex
	VirB1	Peptidoglycan hydrolase, involved in T-DNA transfer
	VirB3	Inner membrane protein forms a channel for the type IV secretion of the T-DNA complex
	VirB6	Inner membrane protein of the type IV secretion of the T-DNA complex
	VirB10	Inner membrane protein of type IV secretion of T-DNA complex, TonB-like
	VirD4	Coupling protein, ATPase required for T-DNA transfer

*: Gene names that belong to the seven cluster groups in the flagella family.

**Table 3 microorganisms-13-02473-t003:** Gene sequence variation analysis study of the SE and SK isolates that could affect the function and performance of some important genes that are associated with biofilm formation, and related components against isolate SE 53932 and SK PW9-3 as reference, respectively.

Serovar	Isolates	Type	Function	Upstream Feature	Downstream Feature	snpEff Type	snpEff Impact
SE	715701 (I), 79801 (I), 79901 (I), 52239 (S), 51094 (S), 51095 (S)	Nonsyn	Cellulose synthase catalytic subunit [UDP-forming] (EC 2.4.1.12)	Cellulose biosynthesis protein BcsQ	Cyclic di-GMP-binding protein BcsB	stop gained	High
SE	715701, 79801, 79901 (I)	Deletion	Cellulose synthase operon protein C	beta-1,4-glucanase (cellulase) (EC 3.2.1.4)	C-di-GMP phosphodiesterase (EC 3.1.4.52)	frameshift	High
SE	715701 (I), 79801 (I), 79901 (I), 52239 (S), 51094 (S), 51095 (S)	Insertion	Uncharacterized fimbrial-like protein YehA	Outer membrane usher protein YehB	Nickel/cobalt homeostasis protein RcnB	frameshift	High
SK	F43-3 (N)	Nonsyn	TolA protein	Tol biopolymer transport system, TolR protein	Tol-Pal system beta propeller repeat protein TolB	missense	Moderate
SE	53936 (S)	Nonsyn	TolA protein	Tol biopolymer transport system, TolR protein	Tol-Pal system beta propeller repeat protein TolB	missense	Moderate
SE	52239, 51094, 53936 (S)	Nonsyn	TolA protein	Tol biopolymer transport system, TolR protein	Tol-Pal system beta propeller repeat protein TolB	missense	Moderate
SE	719001 (S), 715701 (I), 52239 (S), 51094 (S), 710901 (S), 53936 (S), 618901 (S), 619102 (S)	Nonsyn	TolA protein	Tol biopolymer transport system, TolR protein	Tol-Pal system beta propeller repeat protein TolB	missense	Moderate
SE	107 (I)	Nonsyn	UPF0126 inner membrane protein YadS	Vitamin B12 ABC transporter, substrate-binding protein BtuF	Iron-sulfur cluster insertion protein ErpA	missense	Moderate
SE	107 (I)	Nonsyn	Glutamate-1-semialdehyde 2,1-aminomutase (EC 5.4.3.8)	FIG01048481: hypothetical protein	Uncharacterized protein YadU in the stf fimbrial cluster	missense	Moderate
SE	715701, 79801, 79901 (I)	Nonsyn	Glutamate-1-semialdehyde 2,1-aminomutase (EC 5.4.3.8)	FIG01048481: hypothetical protein	Uncharacterized protein YadU in the stf fimbrial cluster	missense	Moderate
SE	715701 (I), 79801 (I), 79901 (I) 52239 (S), 51094 (S), 51095 (S)	Nonsyn	Flagellar basal-body P-ring formation protein FlgA	Negative regulator of flagellin synthesis, FlgM (anti-sigma28)	Flagellar basal-body rod protein FlgB	missense	Moderate
SE	107 (I)	Nonsyn	Cytoplasmic alpha-amylase (EC 3.2.1.1)	Flagellar biosynthesis protein FliT	Uncharacterized lipoprotein YedD	missense	Moderate
SE	52239 (S)	Nonsyn	Curli production assembly/transport component CsgF	Curli production assembly/transport component CsgG	Curli production assembly/transport component CsgE	missense	Moderate
SK	CW8-3 (W), F38-2 (N), F39-3 (N), F41-2 (N), F42-3 (N), F43-3 (N), F44-1 (S), F45-2 (N), W36-3 (S), W37-2 (N), W38-1 (N), W39-1 (N), W41-2 (N), W42-2 (N), W44-3 (N), F36-3 (N)	Synon	TolA protein	Tol biopolymer transport system, TolR protein	Tol-Pal system beta propeller repeat protein TolB	synonymous	Low
SK	C45-1 (N), CW8-3 (W), F38-2 (N), F39-3 (N), F41-2 (N), F42-3 (N), F43-3 (N), F44-1 (S), F45-2, W36-3 (S), W37-2 (S), W38-1 (N), W39-1 (N), W41-2 (N), W44-3 (N), F36-3 (N)	Synon	TolA protein	Tol biopolymer transport system, TolR protein	Tol-Pal system beta propeller repeat protein TolB	synonymous	Low
SK	CW8-3 (W), W38-1 (N), W39-1 (N), W41-2 (N), F36-3 (N)	Synon	TolA protein	Tol biopolymer transport system, TolR protein	Tol-Pal system beta propeller repeat protein TolB	synonymous	Low
SK	W38-1, W39-1 (N)	Synon	TolA protein	Tol biopolymer transport system, TolR protein	Tol-Pal system beta propeller repeat protein TolB	synonymous	Low
SE	715701 (I), 79801 (I), 52239 (S), 51094 (S), 710901 (S), 53936 (S)	Synon	TolA protein	Tol biopolymer transport system, TolR protein	Tol-Pal system beta propeller repeat protein TolB	synonymous	Low
SE	107 (I)	Synon	Cellulose biosynthesis protein BcsG	Small inner membrane protein, YmgF family	Cellulose biosynthesis protein BcsF	synonymous	Low
SE	107 (I)	Synon	Cellulose biosynthesis protein BcsQ	Putative cytoplasmic protein YhjR	Cellulose synthase catalytic subunit [UDP-forming] (EC 2.4.1.12)	synonymous	Low
SE	107 (I)	Synon	Cellulose synthase operon protein C	beta-1,4-glucanase (cellulase) (EC 3.2.1.4)	c-di-GMP phosphodiesterase (EC 3.1.4.52)	synonymous	Low
SE	107 (I)	Synon	Uncharacterized protein YadE	Putative PTS system IIA component YadI	Aspartate 1-decarboxylase (EC 4.1.1.11)	synonymous	Low
SE	107 (I)	Synon	tRNA (5-methylaminomethyl-2-thiouridylate)-methyltransferase (EC 2.1.1.61)/FAD-dependent cmnm(5)s(2)U34 oxidoreductase	Uncharacterized protein YfcL	3-oxoacyl-[acyl-carrier-protein] synthase, KASI (EC 2.3.1.41)	synonymous	Low
SE	107 (I)	Synon	Uncharacterized protein YfcC	Transketolase, C-terminal section (EC 2.2.1.1)	BioD-like N-terminal domain/Phosphate acetyltransferase (EC 2.3.1.8)	synonymous	LOW
SE	107 (I)	Synon	Flagellar motor rotation protein MotA	Flagellar motor rotation protein MotB	Flagellar transcriptional activator FlhC	synonymous	LOW
SE	107 (I)	Synon	hypothetical protein	Flagellar regulon repressor RtsB	hypothetical protein	synonymous	LOW

Synon = synonymous, Nonsyn = nonsynonymous. S = strong, I = intermediate, W = weak, N = non-biofilm producers. snpEFF type = snpEFF variant type, snpEFF impact = snpEFF variant impact.

**Table 4 microorganisms-13-02473-t004:** Protein encoding gene comparison between SE 53932 and SE 107.

Proteins Present in Strong SE 53932, but Absent in Intermediate SE 107 Excluding Hypothetical Proteins and Proteins with Unknown Functions
Exoenzymes regulatory protein AepA precursor
putative e6 protein
ail and ompX Homolog
Phage repressor protein cI
Phage Cox (control of excision) protein
Phage activator protein cII
FIL protein
Phage replication protein GpB *
Phage Orf80 protein *
Phage replication protein GpA, endonuclease *
Phage protein *
DNA-damage-inducible protein I *
Phage protein *
Phage portal vertex protein GpQ *
Phage terminase, ATPase subunit GpP *
Phage capsid scaffolding protein GpO *
Phage major capsid protein GpN *
Phage terminase, endonuclease subunit GpM *
Phage head completion-stabilization protein GpL *
Phage tail protein GpX *
Phage holin *
Phage lysis regulatory protein, LysB *
Phage tail completion protein GpR *, GpS *
Phage baseplate assembly protein GpV *, GpW *, GpJ *
Phage tail formation protein GpI *
putative inner membrane protein *
Phage tail sheath monomer GpFI *
Phage major tail tube protein GpFII *
Phage tail protein GpE *
Phage P2 GpE family protein *
Phage tail protein GpU *
Phage tail formation protein GpD *
Oxaloacetate decarboxylase Na(+) pump, alpha chain (EC 4.1.1.3)
Anaerobic sulfite reductase subunit A
SdiA-regulated putative outer membrane protein SrgB
Putative cytoplasmic protein
Oxaloacetate decarboxylase Na(+) pump, alpha chain (EC 4.1.1.3)
RelB/StbD replicon stabilization protein (antitoxin to RelE/StbE) **
PI protein **
DNA distortion protein 3 **
Cell division protein FtsH (EC 3.4.24.-) **
IncN plasmid KikA protein **
Coupling protein VirD4, ATPase required for T-DNA transfer **
ATPase required for both assembly of type IV secretion complex and secretion of T-DNA complex, VirB11 **
Inner membrane protein of type IV secretion of T-DNA complex, TonB-like, VirB10 **
Forms the bulk of type IV secretion complex that spans outer membrane and periplasm (VirB9) **
putative conjugal transfer protein
Inner membrane protein of type IV secretion of T-DNA complex, VirB6 **
IncQ plasmid conjugative transfer protein TraG **
Minor pilin of type IV secretion complex (VirB5) **
Inner membrane protein forms channel for type IV secretion of T-DNA complex, VirB3/ATPase required for both assembly of type IV secretion complex and secretion of T-DNA complex, VirB4 **
Pilx2 protein **
Peptidoglycan hydrolase VirB1, involved in T-DNA transfer **
IncQ plasmid conjugative transfer DNA nicking endonuclease TraR (pTi VirD2 homolog) **
DNA distortion protein 1 **
putative membrane protein
Phage integrase
E3 ubiquitin-protein ligase SspH2

* = genes associated with the cluster on the bottom of Figure 5. ** = genes associated with the cluster on the top left of Figure 5.

**Table 5 microorganisms-13-02473-t005:** Protein encoding gene comparison.

Proteins Present in Strong SE 53932, but Absent in Intermediate SE 107 Excluding Hypothetical Proteins and Proteins with Unknown Functions
Phage head, head-DNA stabilization protein D *
Pahe protein ^b^
Ail and ompX Homolog ^b^
Protein KdpF ^a^
Oxaloacetate decarboxylase Na(+) pump, alpha chain (EC 4.1.1.3) *
Kappa-fimbriae probable subunit *
Kappa-fimbriae major subunit *
Kappa-fimbriae regulatory protein *
RepFIB replication protein A *
Putative periplasmic protein *
Putative inner membrane protein *
putative invasin *
Transposase *
27.5 kDa virulence protein *
Actin-ADP-ribosyltransferase, toxin SpvB *
Outer membrane protein *
Virulence genes transcriptional activator *
putative phosphoribulokinase/uridine kinase protein *
putative integrase protein *
RlgA *
Putative cytoplasmic protein *
Alpha-helical coiled coil protein *
Chromosome (plasmid) partitioning protein ParA *, ParB
putative ParB-like nuclease *
Plasmid SOS inhibition protein PsiA *, PsiB *
UPF0380 proteins YafZ and homologs *
IncF plasmid conjugative transfer mating signal transduction protein TraM *
IncF plasmid conjugative transfer regulator TraJ *, TraY *
IncF plasmid conjugative transfer pilin protein TraA *
IncF plasmid conjugative transfer pilus assembly protein TraB *, TraE *, TraK *, TraL *
Oxaloacetate decarboxylase Na(+) pump, alpha chain (EC 4.1.1.3)
secreted effector protein *

^a^ = genes only absent in 715701. ^b^ = genes only absent in 79801. * = genes associated with the cluster on the top left circle in Figure 6.

## Data Availability

The raw data supporting the conclusions of this article will be made available by the authors on request.

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
