# Peer review of "Genomic Drivers of Biofilm Formation in *Salmonella* Enteritidis and *S*. Kentucky from Poultry Production"

_microorganisms, 2025, doi:10.3390/microorganisms13112473_

Round 1

Reviewer 1 Report

Comments and Suggestions for Authors

Comments:

The various biofilm assays need to be repeated due to some very high standard deviations, particularly in the data shown in Figures 1 and 2.

The authors should clarify how they classified the biofilms as high, moderate, or low. It is essential to provide a detailed description of the classification process or include a more specific reference for this categorization.

Additionally, what criteria were used to categorize the different SNP impacts into four groups (high, moderate, low, and modifier) and the 12 specific SNP types? This needs to be explained or supported by a recent reference.

The presence of curli must be demonstrated across different assays in the study.

Furthermore, the conclusions are ambiguous and require significant restructuring. The references also need to be updated to include more recent studies and research. Lastly, the manuscript contains several spelling errors that require correction.

Many sections of the manuscript have been generated with artificial intelligence.

Author Response

Reviewer 1.

Comment 1: The various biofilm assays need to be repeated due to some very high standard deviations, particularly in the data shown in Figures 1 and 2.

Response 1: We appreciate the reviewer’s comment. The biofilm assays were conducted using validated and standardized methods. While some data points show relatively high standard deviations, such variability is expected in biological systems, as reported by others. Importantly, the observed variation did not alter the overall classification or trends in biofilm formation, which remained consistent across experiments. Therefore, we believe that repeating the assays is not needed.

Comment 2: The authors should clarify how they classified the biofilms as high, moderate, or low. It is essential to provide a detailed description of the classification process or include a more specific reference for this categorization.

Response 2: The classification of the biofilms was described under section 2.1.3., second paragraph from line 184 to 187, with a specific reference for this categorization.

 Comment 3: Additionally, what criteria were used to categorize the different SNP impacts into four groups (high, moderate, low, and modifier) and the 12 specific SNP types? This needs to be explained or supported by a recent reference.

Response 3: Thank you for the comment. The BV-BRC pipeline uses a computer program called SnpEff to categorize the effects of variants in genome sequences. The BV-BRC website and the table obtained from BV-BRC below explain this as follows “SnpEff rapidly categorizes the effects of variants in genome sequences. Once a genome is sequenced, SnpEff annotates variants based on their genomic locations and predicts coding effects. Annotated genomic locations include intronic, untranslated region, upstream, downstream, splice site, or intergenic regions. Coding effects such as synonymous or non-synonymous amino acid replacement, start codon gains or losses, stop codon gains or losses, or frame shifts can be predicted. The BV-BRC variant analysis service provides a var.snpEFF.vcf file for each of the read libraries that were loaded” (Variation - SNP Analysis Service | BV-BRC). A clarification was added under section 2.2.3. Line 261- 264.   “SNP impacts were categorized using SnpEff, a variant annotation tool integrated into the BV-BRC pipeline. SnpEff predicts the functional consequences of each SNP based on its genomic context and classifies them into four categories, high, moderate, low, and modifier, according to the expected effect on gene or protein function”

 Comment 4: The presence of curli must be demonstrated across different assays in the study.

Response 4: The assays used to evaluate curli and cellulose production in this study are well-established and widely applied in biofilm research. Although curli expression could also be assessed by PCR, such molecular confirmation would not add meaningful information in this context, as the genomic data for all strains are already available and confirm the presence of the corresponding genes. Our objective was to assess the phenotypic expression of curli, and the selected assay is a validated and reliable method for this purpose.

Comments 5: Furthermore, the conclusions are ambiguous and require significant restructuring. The references also need to be updated to include more recent studies and research. Lastly, the manuscript contains several spelling errors that require correction.

Response 5: While we respectfully disagree that the original conclusions were ambiguous, we acknowledge that the study involves multiple layers of analysis and have made further revisions to improve clarity and readability. The manuscript has been thoroughly proofread to correct spelling and typo errors.

Comment 6: Many sections of the manuscript have been generated with artificial intelligence.

Response 6: AI was only used for the purpose of improving English grammar (Grammarly). No text was generated using AI.  

Reviewer 2 Report

Comments and Suggestions for Authors

The paper by Zhang et al. investigates the biofilm formation ability of Salmonella Enteritidis (SE) and S. Kentucky (SK) from poultry production and its genetic determinants. The manuscript is well written, the scientific methods are sound, and the findings are of great interest to the scientific community, given the growing attention toward biofilm-forming bacteria. I have only a few minor comments that I would like the authors to address before publication:

  • Lines 230–232: This sentence should be moved to the Results section. In addition, the authors should provide more detailed information on the raw read statistics, including the number of reads, total base pairs, and sequencing coverage for each individual strain. I suggest reporting the mean values with standard deviations in the main text and including a supplementary file with the detailed data.

  • Statistical analysis: Could the authors provide more details regarding the statistical analysis performed? Why was the Chi-square test chosen, and which variables were considered? For instance, what do the authors mean by “interactions between isolates and biofilm categories”?

  • Figure 4: The figure is difficult to read and should be improved, as it presents some of the key findings of the study.

Author Response

We sincerely thank the reviewers for their time in reviewing our manuscript and providing valuable feedback.

Reviewer 2

The paper by Zhang et al. investigates the biofilm formation ability of Salmonella Enteritidis (SE) and S. Kentucky (SK) from poultry production and its genetic determinants. The manuscript is well written, the scientific methods are sound, and the findings are of great interest to the scientific community, given the growing attention toward biofilm-forming bacteria. I have only a few minor comments that I would like the authors to address before publication:

Comment 1: Lines 230–232: This sentence should be moved to the Results section. In addition, the authors should provide more detailed information on the raw read statistics, including the number of reads, total base pairs, and sequencing coverage for each individual strain. I suggest reporting the mean values with standard deviations in the main text and including a supplementary file with the detailed data.

Response 1. Thank you for the comment. We retained the pangenome information in the Methods section, as it provides essential context for explaining the analytical approach. A table containing the requested information has been added to the supplementary materials.

Comment 2: Statistical analysis: Could the authors provide more details regarding the statistical analysis performed? Why was the Chi-square test chosen, and which variables were considered? For instance, what do the authors mean by “interactions between isolates and biofilm categories”?

Response 2: Thank you for the comment. Additional details have been added to clarify the statistical analysis. The Chi-square test was selected because the variables analyzed, biofilm formation categories (strong, moderate, weak, and non-producer) and the expression of curli, cellulose, and specific SNP types, are categorical in nature. It specifically assesses associations between qualitative variables. The phrase “interactions between isolates and biofilm categories” refers to testing whether the distribution of isolates across biofilm categories was independent of genomic characteristics (e.g., SNP type or predicted impact). In other words, we evaluated whether specific genotypic traits (such as SNP impact levels) were significantly associated with particular phenotypic outcomes (biofilm strength, curli, and cellulose production). Statistical significance was determined at p < 0.0001. A clarification was added

Line 283-288: Associations between phenotypic traits, including biofilm formation category (strong, moderate, weak, and non-producer), and the production of curli and cellulose and genomic characteristics (SNP type and predicted SNP impact) were analyzed using the Chi-square test. Specifically, the analysis evaluated whether the distribution of iso-lates across biofilm formation categories was independent of genomic traits, such as the presence or absence of specific SNPs or their functional impact. Statistical significance was determined at p < 0.0001.

Comment 3: Figure 4: The figure is difficult to read and should be improved, as it presents some of the key findings of the study.

Response 3: Thank you for the comment. We have revised Figure 4 and updated the figure legend to provide a more detailed description of each element represented, including the meaning of the colored branches, strain sources, biofilm categories, and the presence or absence of key genes.

Reviewer 3 Report

Comments and Suggestions for Authors

This is a very interesting study investgating the genomic drivers of biofilm formation in Salmonella enteritidis and Salmonella Kentucky from poultry production. The findings demostrate that genetic variation, not just gene presence, shaped biofilm phenotypes. The study was well-designed and written in good English. The results were clear and well-supported the conclusion. Overall, the current manuscript could be considered for publication after only minor revisions. 

  1. The authors should clarify why SE and SK strains were not obtained from national microorganism colllection center.
  2. P180-181, References were needed to describe the methods provided by Stepanovic et al. and Adator et al..
  3. 2.2.1 and 2.2.2 could be combined in one paragraph.
  4. For figure 1 and figure 2, add the repeat of number.
  5. In the conclusion part, the authors should state that further experiments were needed to verify the role of tolA in determing biofilm formation ability of different SE isolates. 

Author Response

We sincerely thank the reviewers for their time in reviewing our manuscript and providing valuable feedback

Reviewer 3

This is a very interesting study investgating the genomic drivers of biofilm formation in Salmonella enteritidis and Salmonella Kentucky from poultry production. The findings demostrate that genetic variation, not just gene presence, shaped biofilm phenotypes. The study was well-designed and written in good English. The results were clear and well-supported the conclusion. Overall, the current manuscript could be considered for publication after only minor revisions. 

Comment 1: The authors should clarify why SE and SK strains were not obtained from national microorganism colllection center.

Response 1: Thank you for the comment. We did not include isolates from the National Microorganism Collection Center or ATCC because our objective was to compare wild-type isolates obtained directly from poultry facilities. Although ATCC reference strains were used as quality controls in our assays, they were not included in the genomic or phenotypic analyses, as our focus was on environmental and production-related isolates that better represent poultry production.

Comment 2: P180-181, References were needed to describe the methods provided by Stepanovic et al. and Adator et al..

Response 2: Thank you for the comment. The references for both Stepanović et al. and Adator et al. were already included at the end of the paragraph; however, we have now repositioned them closer to the authors’ names to improve clarity and proper citation format.

Comment 3: 2.2.1 and 2.2.2 could be combined in one paragraph.

Response 3: Agree. We have combined those two sections into one.

Comment 4: For figure 1 and figure 2, add the repeat of number.

Response 4: Agree. Thank you for pointing it out. We added this information in the captions of figure 1 and figure 2.

Comment 5: In the conclusion part, the authors should state that further experiments were needed to verify the role of tolA in determing biofilm formation ability of different SE isolates. 

Response 5: We agree with the reviewer’s suggestion. A statement has been added to the conclusion.

 Line 797-802: Variations in tolA, tolB, and tolR genes, particularly nonsynonymous mutations in tolA, may influence protein function and reduce biofilm-forming capacity, with Salmonella Enteritidis isolates showing more high-impact mutations than S. Kentucky. These findings identify tolA as a potential determinant of biofilm variability; however, further functional studies are needed to verify its specific role in regulating biofilm formation across different S. Enteritidis isolates.